# Synthesis of structurally controlled hyperbranched polymers using a monomer having hierarchical reactivity

Yangtian Lu[1], Takashi Nemoto[1], Masatoshi Tosaka[1] & Shigeru Yamago [1]

Hyperbranched polymers (HBPs) have attracted significant attention because of their characteristic topological structure associated with their unique physical properties compared with those of the corresponding linear polymers. Dendrimers are the most structurally controlled HBPs, but the necessity of a stepwise synthesis significantly limits their applications in materials science. Several methods have been developed to synthesize HBPs by a one-step procedure, as exemplified by the use of $AB_2$ monomers and AB′ inimers under condensation and self-condensing vinyl polymerization conditions. However, none of these methods provides structurally controlled HBPs over the three-dimensional (3D) structure, i.e., molecular weight, dispersity, number of branching points, branching density, and chain-end functionalities, except under special conditions. Here, we introduce a monomer design concept involving two functional groups with hierarchical reactivity and demonstrate the controlled synthesis of dendritic HBPs over the 3D structure by the copolymerization of the designed monomer and acrylates under living radical polymerization conditions.

[1] Institute for Chemical Research Kyoto University, Uji, 611-0011, Japan. Correspondence and requests for materials should be addressed to S.Y. (email: yamago@scl.kyoto-u.ac.jp)

Polymer materials are indispensable to our modern society and enrich the quality of our lives. While linear polymers are almost entirely used for fabricating polymer materials, hyperbranched polymers (HBPs) have attracted significant attention due to their advantageous physical properties compared to those of their linear analogs, such as lower intrinsic viscosity, low glass transition temperature, and a large number of terminal groups[1–5]. Therefore, HBPs are promising in many applications, e.g., as lubricants, coatings, drug-delivery vehicles, and catalysts[6–10]. Control over the three-dimensional (3D) structure of HBPs, i.e., molecular weight, dispersity, number of branching points, branching density, and chain-end functionalities, would significantly improve and modify the polymer properties and contribute to the design and synthesis of new polymer materials. However, there is no practical or effective method to synthesize HBPs with well-controlled structures. The situation is in sharp contrast to that of linear polymers, in which living polymerization methods have been developed and are used in industry.

Dendrimers and dendrons possess the most controlled structure over the 3D shape, but the requirement of multistep synthesis limits their availability[11–15]. The polycondensation of AB$_2$ monomers[16,17], in which A and B refer to functional groups that react with each other, and self-condensing vinyl polymerization

(SCVP)[18] or self-condensing vinyl copolymerization (SCVCP) using AB* monomer 1[19], in which a vinyl group serves as the A group and B* is an initiator group, represent more accessible synthetic routes to HBPs (Fig. 1a). However, control of the corresponding 3D macromolecular structure is limited, and polymers with polydisperse and irregularly branched structures are obtained. This is due to the presence of a step-growth mechanism, and monomer 1 as well as all existing oligomers, such as 2 and 3, randomly react with the growing polymer chains and each other. Therefore, the simple application of living polymerization to HBP synthesis by SCVP and SCVCP has so far been unsuccessful in achieving structural control under cationic[18], anionic[20,21], radical[22–26], and group transfer polymerization conditions[27,28]. HBPs with narrow dispersity were only obtained at low monomer conversion[23], under specific conditions such as emulsion polymerization to avoid intermicellar reactions[29] or the slow addition technique to avoid the step-growth mechanism[30,31], or through the use of monomers with specific functional groups[32,33]. HBPs with a 100% degree of branching have been achieved, but only under specific reaction conditions[34,35] and without any molecular weight or distribution control.

The poor structural control exhibited by methods using AB$_2$ and AB* monomers results from the presence of a step-growth

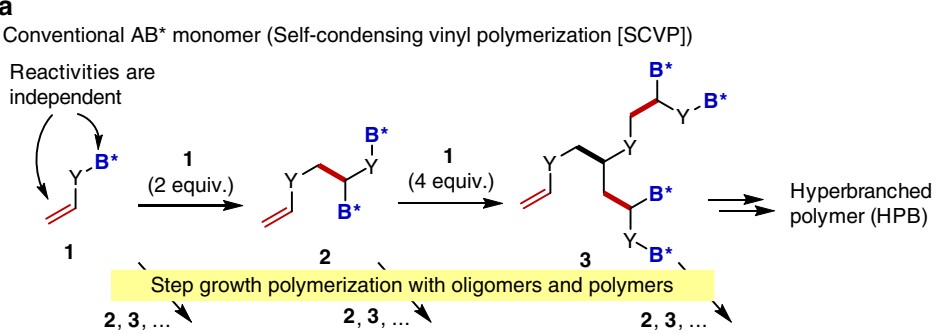

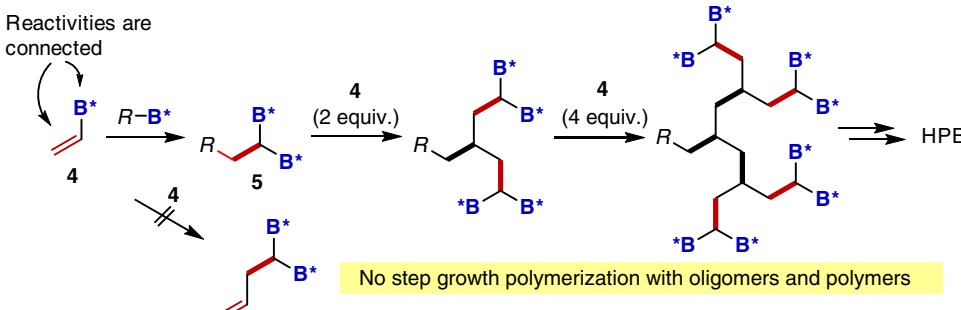

**Fig. 1** Synthetic strategy of hyperbranched polymers (HBPs). **a** The AB* monomer method (self-condensing vinyl polymerization (SCVP)) and **b** a method using an AB* monomer with hierarchical reactivity. The olefin acts as the A functional group, and the bonds originated from the olefin in each step are indicated in red for **a**, **b**. **c** Vinyl telluride design. Carbon–tellurium bond dissociation energy (BDE) in kJ mol$^{-1}$ obtained by density functional theory calculations at the (U)B3LYP/6-31 G(d,p)(C,H) + LANL2DZ(Te) level

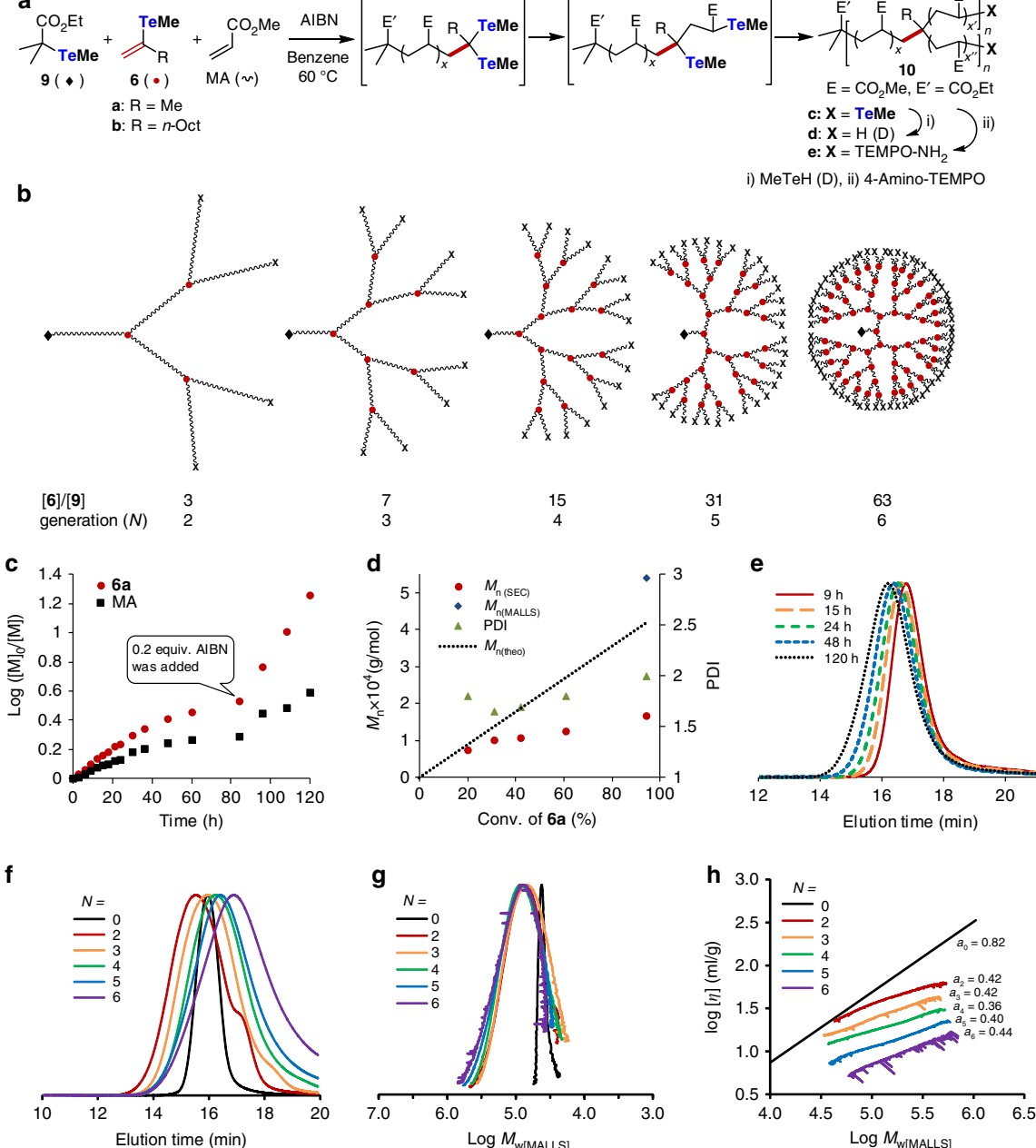

**Fig. 2** Synthesis and characterization of dendritic hyperbranched polymers (HBPs). **a** Formation of the HBPs by the copolymerization of **6** and methyl acrylate (MA) in the presence of an organotellurium chain-transfer agent **9**. **b** Schematic structures of ideal polymer products produced at [**6**]/[**9**] ratios of 3, 7, 15, 31, and 63, corresponding to dendritic generations $N$ of 2, 3, 4, 5, and 6, respectively. **c** Time evolution of the consumption of **6a** and MA determined by $^1$H NMR analysis for the synthesis of the sixth generation (Table 1, run 5). Additional AIBN (0.2 equiv.) was added after 84 h. **d** Correlation among the monomer conversion, number average molecular weight, and PDI for the synthesis of the sixth generation (Table 1, run 5). **e** Time evolution of the SEC traces from 9 to 120 h. **f** SEC traces observed by a refractive index (RI) detector. **g** Corrected SEC traces with the weight average molecular weight determined by MALLS ($M_{w[MALLS]}$) and peak intensity determined by an RI detector. **h** Mark–Houwink–Kuhn–Sakurada plot for linear PMA (dendritic generation $N = 0$) and copolymers with $N = 2$, 3, 4, 5, and 6 (Table 1, runs 1–6)

mechanism arising from the independent reactivity of A and B/B* groups. Therefore, we envisioned that the polymerization could be controlled if the two groups of a monomer have hierarchical reactivity, e.g., if B* only participates in polymerization after A has reacted (Fig. 1b). Then, the repetition of this process under living polymerization conditions would give HBPs with controlled 3D structures, since the step-growth mechanism would no longer be involved.

Here, we report the design of such a monomer and its copolymerization under living radical polymerization conditions (also known as reversible-deactivation radical polymerization). This concept should be applicable to all living polymerization methods, but organotellurium-mediated radical polymerization[36, 37] was used here as proof of principle due to its high synthetic versatility. We selected a vinyl telluride as a monomer having hierarchical reactivity, which does not initiate polymerization by itself but instead is involved at the start of the polymerization and acts as a branching point after it reacts. Copolymerization of the vinyl telluride and acrylates in the presence of an organotellurium chain-transfer agent (CTA) gives dendritic HBPs with controlled

**Table 1 Synthesis of branched polymers by the copolymerization of 6a and acrylate monomers via organotellurium-mediated radical polymerization[a]**

| Run | [9]/[6a]/[MA] (generation N) | Time (h) | Conv. (%) | | $M_{n[theo]}$ ($\times 10^4$ g mol$^{-1}$) | $M_{n[SEC]}$[a] ($\times 10^4$ g mol$^{-1}$) | PDI[a] | $M_{n[MALLS]}$[b] ($\times 10^4$ g mol$^{-1}$) | $\overline{X_{br}}$[c] |
|---|---|---|---|---|---|---|---|---|---|
| | | | 6a | MA | | | | | |
| 1 | 1/3/500 (2) | 24 | >99 | 95 | 4.12 | 4.21 | 1.55 | 5.37 | 68 |
| 2 | 1/7/500 (3) | 30 | >99 | 94 | 4.10 | 3.27 | 1.62 | 5.35 | 31 |
| 3 | 1/15/500 (4) | 43 | >99 | 90 | 3.96 | 2.56 | 1.71 | 5.73 | 16 |
| 4[d] | 1/31/500 (5) | 84 | >99 | 90 | 4.03 | 1.71 | 1.97 | 5.60 | 7.1 |
| 5[e] | 1/63/500 (6) | 120 | 95 | 74 | 3.46 | 0.98 | 1.99 | 5.39 | 2.9 |
| 6 | 1/0/500 (0) | 2 | – | 90 | 3.90 | 3.96 | 1.12 | 3.72 | – |
| 7 | 1/15/100 (4) | 66 | >99 | 92 | 0.86 | 0.56 | 1.40 | 0.91 | 3.0 |
| 8 | 1/15/250 (4) | 66 | >99 | 94 | 2.07 | 1.20 | 1.87 | 2.40 | 7.6 |
| 9 | 1/15/2000 (4) | 80 | >99 | 93 | 16.08 | 7.47 | 1.91 | 16.22 | 60 |
| 10[f] | 1/127/2000 (7) | 132 | >99 | 90 | 16.01 | 5.69 | 2.08 | 17.65 | 7.3 |
| 11 | 1/15[g]/500 (4) | 72 | >99 | 84 | 3.85 | 2.18 | 1.80 | ND[h] | 14 |
| 12 | 1/15[g]/500 (4) | 72 | >99 | 85 | 3.89 | 1.89 | 1.73 | ND[h] | 14 |
| 13 | 1[i]/15/500 (4) | 43 | >99 | 92 | 4.04 | 2.95 | 1.49 | 5.42 | 15 |
| 14[j] | 1[i]/30/500 (4) | 80 | >99 | 97 | 4.32 | 1.74 | 1.89 | 4.67 | 7.8 |
| 15[k] | 1[i]/21/500 (3) | 168 | >99 | 84 | 3.73 | 1.73 | 1.44 | 5.66 | 9.3 |
| 16[l] | 1/15/500 (4) | 60 | >99 | 80 | 3.53 | 1.34 | 2.12 | 4.52 | 13 |
| 17[k] | 1/15/500[m] (4) | 84 | >99 | 71 | 5.17 | 2.51 | 1.85 | ND[h] | 11 |
| 18 | 1/15/500[m] (4) | 7 | 81 | 86 | 4.34 | 2.02 | 1.45 | 3.60[n] | 14 |

AIBN azobisisobutyronitrile, $^1$H NMR proton nuclear magnetic resonance, MA methyl acrylate, MALLS multi-angle laser light scattering, ND not determined, PDI polydispersity index, SEC size-exclusion chromatography, 4-amino-TEMPO 4-Amino-2,2,6,6-tetramethylpiperidine-1-oxyl
[a]Determined by SEC calibrated against poly(methyl methacrylate) standards
[b]The weight average molecular weight obtained by MALLS ($M_{w[MALLS]}$) was divided by the PDI ($M_n/M_w$) obtained by SEC
[c]Average number of monomer units inserted between the branched points calculated by eq. 1. The conversion of MA was used as Conv
[d]Additional AIBN (0.2 equiv.) was added after 24 h
[e]Additional AIBN (0.2 equiv.) was added after 84 h
[f]Additional AIBN (0.2 equiv.) was added after 11 h and 71 h
[g]6b and 6b* were used instead of 6a for runs 9 and 10, respectively
[h]Not determined
[i]11, 12, and 13 were used instead of 9 for runs 13, 14, and 15, respectively
[j]Additional AIBN (0.2 equiv.) was added after 12 h
[k]Additional AIBN (0.2 equiv.) was added after 72 h
[l]4-amino-TEMPO (1.25 equiv.) was added, followed by irradiation (500 W mercury lamp with a 390 nm cutoff filter) at 25 °C for 6 h
[m]2-(Dimethylamino)ethyl acrylate and N,N-dimethylacrylamide were used instead of MA for runs 16 and 17, respectively
[n]Determined from $^1$H NMR

3D structures in one step. The molecular weight, number of branching points, and branching density are easily controlled by changing the relative amounts of CTA, vinyl telluride, and acrylate monomers and the narrow dispersity is maintained. Furthermore, selective transformations of the living polymer-end enable the introduction of functional groups at the chain ends. These results open possibilities for materials design based on HBPs.

## Results

**Design of a monomer having hierarchical reactivity**. We focused on monomer **4**, in which the B* group is directly connected to the vinyl moiety (Fig. 1b); B* in **4** would be inert under radical or cationic polymerization, because vinyl radicals and cations are unstable. However, once **4** is converted to **5** by reaction with an initiator or the dormant species R-B*, both B* groups in **5** become capable of initiating polymerization, because a stable alkyl radical or cation can now be generated.

The reactivity of **4** and the dormant species of the propagating polymer ends, shown in Fig. 2a, were estimated by density functional theory calculations at the B3LYP/6-31 G(d,p)(C,H) + LAN2DZ(Te) level (Fig. 1c, Supplementary Note 1, and Supplementary Tables 1 and 2). The carbon–tellurium bond dissociation energy (BDE) of vinyl telluride **6a** (R = Me) was calculated to be 216 kJ mol$^{-1}$ (Fig. 1c), and this value is significantly higher than those of **7** (160 kJ mol$^{-1}$) and **8** (154 kJ mol$^{-1}$), which are model compounds of the propagating polymer ends. This difference strongly suggests the selective activation of **7** and **8** in preference to **6a** under polymerization conditions. Furthermore, the similar BDEs of **7** and **8** ensure the

simultaneous activation of the two tellurium groups in **7** once it forms.

**Synthesis and characterization of hyperbranched PMA**. **6a** was easily prepared by modifying a previously reported procedure (Supplementary Notes 2, 3, and 4 and Supplementary Figs. 19 and 20)[38, 39] and copolymerized with methyl acrylate (MA, 500 equivalents (equiv.)) in the presence of organotellurium CTA **9**[12] at different [**6a**]/[**9**] ratios of 3, 7, 15, 31, and 63 in the presence of azobisisobutyronitrile (AIBN) (0.2 equiv.) as a radical initiator at 60 °C (Fig. 2a, Table 1, runs 1–5, and Supplementary Notes 6–10)[40]. When all dormant species corresponding to **7** and **8** have equal reactivity, polymerization should yield dendritic HBPs corresponding to the second to sixth dendritic generations (N = 2–6) depending on the [**6a**]/[**9**] ratio, in which the initiating group derived from **9** and branching points derived from **6a** are connected by a poly (methyl acrylate) (PMA) chain. Generation N leads to $(2^N - 1)$ branching points and $[2^{(N + 1)} - 1]$ branched polymer chains (Fig. 2b). The average number of monomer units ($\overline{X_{br}}$) inserted between the branched points can be estimated as

$$\overline{X_{br}} = \frac{[MA]}{[9][2^{N+1} - 1]} \times Conv \qquad (1)$$

under ideal statistical copolymerization conditions, in which Conv refers to the conversion of the monomer.

Monitoring polymerization by proton nuclear magnetic resonance ($^1$H NMR) and size-exclusion chromatography (SEC) revealed that **6a** was quantitatively converted into the copolymer with high MA conversion in all cases, though the addition of **6a**

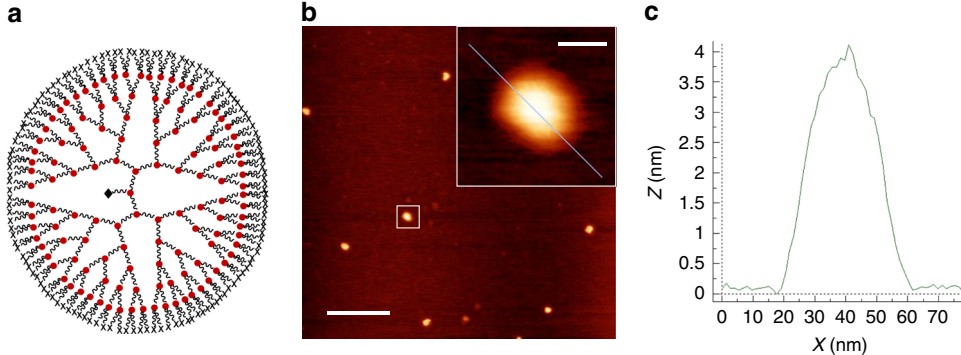

**Fig. 3** Structure and microscopy images of the seventh generation dendritic hyperbranched polymer. **a** Schematic structures of the ideal polymer product, **b** height image (for the magnified image, see inset). Scale bar: 200 and 20 nm (for the magnified image), and **c** cross-sectional profile obtained by AFM for a sample prepared by spin-casting onto a freshly cleaved mica surface with a solution of **10d** (0.001 mg/ml in CHCl₃) prepared at a [**6**]/[**9**] ratio of 127, which corresponds to a dendritic generation $N$ of 7 (Table 1, run 10)

significantly slowed the rate of polymerization compared to that when **6a** was absent (run 6). This is probably due to the lower activation efficiency of the dormant species corresponding to **7** and **8** than that of the acrylate end having a BDE of approximately 140 kJ mol⁻¹. Fig. 2c–e and Supplementary Fig. 27 show the results of polymerization monitoring for $N = 6$ ([**6a**]/[**9**] = 63, run 5) corresponding to the sixth dendritic generation. The consumption of **6a** and MA followed pseudo-first-order kinetics at the beginning of the polymerization, indicating the occurrence of controlled polymerization, while **6a** was consumed slightly faster than MA (Fig. 2c). The polymerization rate gradually decreased with increasing monomer conversion. Therefore, an additional 0.2 equiv. of AIBN was added after 84 h, which significantly increased the reaction rate. Despite the change in the reaction rate, the relative monomer consumption (**6a**/MA) was constant, indicating the formation of a statistical copolymer.

SEC analyses were carried out after reduction of the methyltellanyl polymer-end group in **10c** (X = TeMe) to **10d** (X = H) by methyltellanol (MeTeH), which was generated in situ from MeTeSiMe₃ (Supplementary Note 5) and methanol[41]. The number average molecular weight determined by SEC ($M_{n[SEC]}$) using poly(methyl methacrylate) (PMMA) standards increased with increasing monomer conversion; $M_{n[SEC]}$ showed good agreement with the theoretical values ($M_{n[theo]}$) only at low monomer conversion (< 20%) and deviated significantly as the monomer conversion increased (Fig. 2d). These results are consistent with the formation of a branched polymer, which has a smaller hydrodynamic volume than linear polymers[1]. At 95% and 74% conversion of **6a** and MA, respectively, $M_{n[SEC]}$ ($0.98 \times 10^4$) was significantly smaller than $M_{n[theo]}$ ($3.46 \times 10^4$). In contrast, the $M_n$ determined by multi-angle laser light scattering (MALLS, $M_{n[MALLS]}$) was $5.39 \times 10^4$ (Supplementary Fig. 1), which is close to $M_{n[theo]}$ ($M_{n[MALLS]}$ was determined from weight average molecular weight obtained from MALLS ($M_{w[MALLS]}$) divided by the polydispersity index (PDI) ($M_n/M_w$) obtained by SEC. This is because PDI determined by MALLS is low reliability due to the weak intensity of low-molecular weight part as the targeted molecular weight is moderate. $M_{w(MALLS)}$ and $M_{n(MALLS)}$ would be also overestimated by the same reason.). The PDI remained below 2.0 throughout the polymerization, which is significantly lower than the PDI obtained by SCVP and SCVCP. In addition, the SEC trace was unimodal throughout the polymerization even at high monomer conversion (Fig. 2e). All results are consistent with the formation of HBPs with well-defined structures.

The generality of this method was supported by the controlled synthesis of HBPs with different dendritic generations ($N = 2$–6) by changing the [**6a**]/[**9**] ratio from 3 to 63 while keeping the [**9**]/

[MA] ratio constant (runs 1–5). All copolymerization reactions afforded copolymers with unimodal distribution, except for $N = 2$, in which a small shoulder was observed in the low-molecular-weight region due to the formation of a PMA homopolymer (Fig. 2f). As the current method depends on statistical copolymerization between **6a** and MA, a homopolymer inevitably forms for low dendritic generations. However, for $N > 3$, unimodal SEC traces were observed in all cases. Despite the appearance of a shoulder for $N = 2$, the PDI was below 2.0. The $M_{n[SEC]}$s calibrated using linear PMMA standards decreased with increasing dendritic generation despite the similarity of the $M_{n[theo]}$s. In contrast, the $M_{n[MALLS]}$s remained almost unchanged, close to the $M_{n[theo]}$s (Fig. 2g).

The intrinsic viscosity ([η]) of the HBPs estimated from the Mark–Houwink–Kuhn–Sakurada (MHKS) plot was smaller than that of linear PMA of equal $M_w$ in all samples (Fig. 2h and Supplementary Figs. 15 and 16). Furthermore, the viscosity of the HBPs at the same $M_{w(MALLS)}$ decreased as the branching number increased. The MHKS exponent α obtained from the slope is 0.36–0.44, which is significantly smaller than that of linear PMA (0.82). All these results support the formation of the desired HBPs.

Control of the branching density and molecular weight, i.e., control of the PMA spacer length between branching points, was demonstrated by changing the MA/**9** ratio to 100, 250, and 2000 while maintaining the **6a**/**9** ratio at 15 ($N = 4$, Table 1, runs 7–9 and Supplementary Notes 12–14). Quantitative and > 92% conversion of **6a** and MA were observed, respectively, and copolymers showing unimodal SEC traces were obtained in all cases (Supplementary Figs. 2–4). In addition, the $M_{n(SEC)}$s were smaller than the $M_{n(theo)}$s, whereas the $M_{n(MALLS)}$s were close to the $M_{n(theo)}$s in both cases.

An image of a single polymer molecule was directly observed by atomic force microscopy (AFM) for a polymer sample prepared from 125 and 2000 equiv. of **6a** and MA, respectively, relative to **9** ($N = 7$), which is the highest branched polymer prepared in this study (Fig. 3a and Supplementary Note 15). Many spherical dots corresponding to single polymer molecules were observed (Fig. 3b), and this globular shape further supports the formation of HBPs. The typical height and width of the polymer dot was approximately 4 and 40 nm, respectively, suggesting that the HBPs are flattened on the mica surface (Fig. 3c).

**Structural characterization by isotope labeling experiments.** To clarify the branched structure and branching efficiency at the molecular level, the polymer-end structure was characterized by a

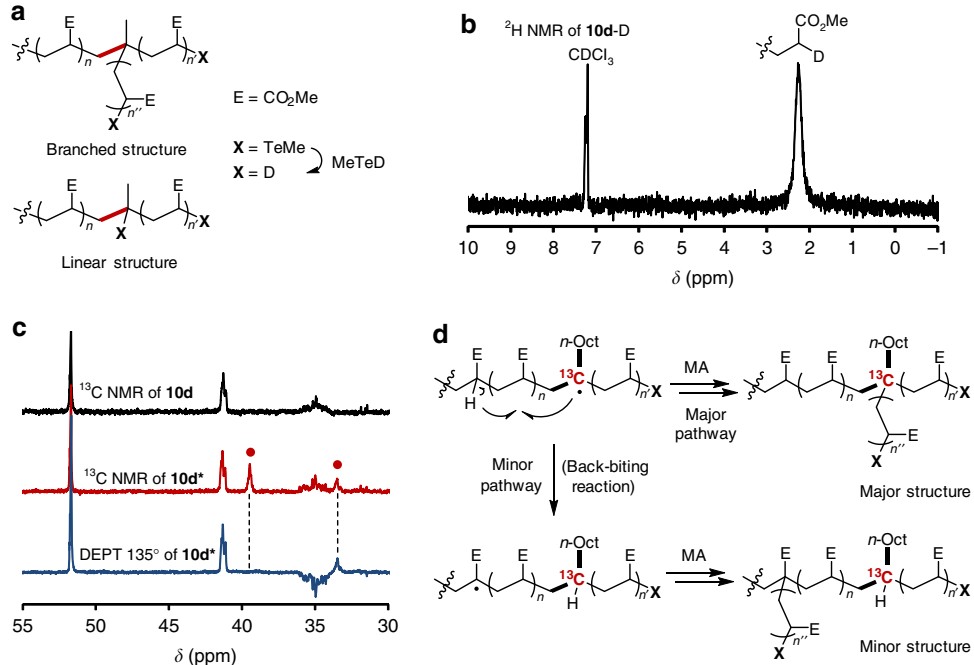

**Fig. 4** Structural analyses of branching by deuterium and $^{13}$C-labeling experiments. **a** Illustration of the linear and branched structures of **10**. **b** $^{2}$H NMR of **10d**-D. **c** $^{13}$C NMR and DEPT 135° of **10d** and **10d\***. Newly observed signals are highlighted as red circles. **d** The major and minor branched structures and their formation mechanism

deuterium labeling experiment. In the branched structure, all polymer-end groups should correspond to acrylate moieties, while both acrylate and alkyl ends should be observed if branching does not occur (Fig. 4a). After copolymerization under similar conditions to those given in Table 1, run 4 ([**9**]/[**6a**]/[MA] = 1/31/500), the polymer-end groups were reduced by deuterated methyltellanol (MeTeD) prepared from MeTeSiMe$_3$ and deuterated methanol (CH$_3$OD), giving **10d**-D (Supplementary Note 24). The $^{2}$H NMR spectrum of the polymer showed a single peak at 2.3 ppm, corresponding to acrylate chain ends, while no signal corresponding to alkyl ends was observed (Fig. 4b). The branching efficiency was determined to be > 99%.

The role of vinyl telluride was further investigated with the help of $^{13}$C-labeled **6b\***, in which the quaternary carbon was selectively labeled by $^{13}$C. Labeled and naturally abundant **6b\*** and **6b** (Supplementary Notes 25–32 and Supplementary Figs. 14, 21–26), respectively, were copolymerized under the same conditions used for **6a**, with [**9**]/[**6b**$^{(*)}$]/[MA] = 1/15/500 (runs 11 and 12 and Supplementary Notes 16 and 17), and the polymer-end groups were reduced by MeTeH. Nearly identical results for the $M_n$ and PDI were obtained regardless of the $^{13}$C labeling. In the $^{13}$C NMR spectrum of the $^{13}$C-labeled polymer, a new peak at 39.4 ppm was observed as the major signal, along with a small peak at 33.4 ppm (Fig. 4c). The major peak at 39.4 ppm was attributed to the quaternary carbon from its disappearance in a distortionless enhancement of the polarization transfer (DEPT) 135° spectrum. The results clearly reveal that vinyl telluride indeed serves as a branching point (Fig. 4d).

The minor signal at 33.4 ppm was assigned to tertiary carbon by the DEPT spectrum. The hydrogen atom at the tertiary carbon most likely originates from the back-biting reaction[42, 43] (Fig. 4d). To verify this possibility, the same $^{13}$C-labeled copolymer was reduced by MeTeD, and the resulting doubly labeled polymer was analyzed by $^{13}$C and $^{2}$H NMR (Supplementary Figs. 17 and 18). The $^{13}$C NMR spectrum was identical to that of the non-deuterated $^{13}$C-labeled sample, and $^{2}$H NMR only showed the acrylate ends. All these results are consistent with the occurrence

of the back-biting reaction. Branching should occur from the resulting mid-chain radical, as its structure is similar to that of polymethacrylate chain-end radicals[44]. All these results indicate that vinyl telluride generates branching points with 100% efficiency.

**Expansion of the synthetic scope.** To further illustrate the synthetic versatility of the current method, HBPs with different structures were synthesized. For example, the copolymerization of **6a** and MA starting from PMA **11** ($M_{n(SEC)} = 1.51 \times 10^4$, PDI = 1.12, [**11**]/[**6a**]/[MA] = 1/15/340) afforded linear-*block*-hyperbranched PMA (run 13 and Supplementary Note 18). Furthermore, the same copolymerization utilizing bifunctional and trifunctional initiators **12** and **13**[45] resulted in dumbbell- and clover-shaped HBPs, respectively (runs 14 and 15 and Supplementary Notes 19 and 20). All polymers exhibited unimodal SEC traces, narrow PDIs, and $M_{n(MALLS)}$s close to the $M_{n(theo)}$s, while the $M_{n(SEC)}$s were significantly smaller than the $M_{n(theo)}$s (Supplementary Figs. 8–10). All these results clearly reveal the formation of HBPs with well-controlled structures and demonstrate the synthetic versatility of the current method (Fig. 5).

As the current method relies on a living polymerization, transformation of the living end is possible. For example, the tellurium end group was quantitatively transformed into an amino group by a radical trapping reaction with 4-amino-2,2,6,6-tetramethylpiperidine-1-oxyl, giving **10e** (run 16, Supplementary Note 23, and Supplementary Fig. 28). Furthermore, monomers having polar functional groups, such as 2-(dimethylamino)ethyl acrylate and *N,N*-dimethylacrylamide, were used as a comonomer instead of MA, as radical polymerization is highly compatible with polar functional groups. The corresponding HBPs with controlled molecular structures having narrow PDIs were synthesized in both cases (runs 17 and 18, Supplementary Notes 21 and 22, and Supplementary Figs. 11–13). A wide variety of HBPs having different functional groups can be synthesized by changing the comonomer and the method of end-group transformation.

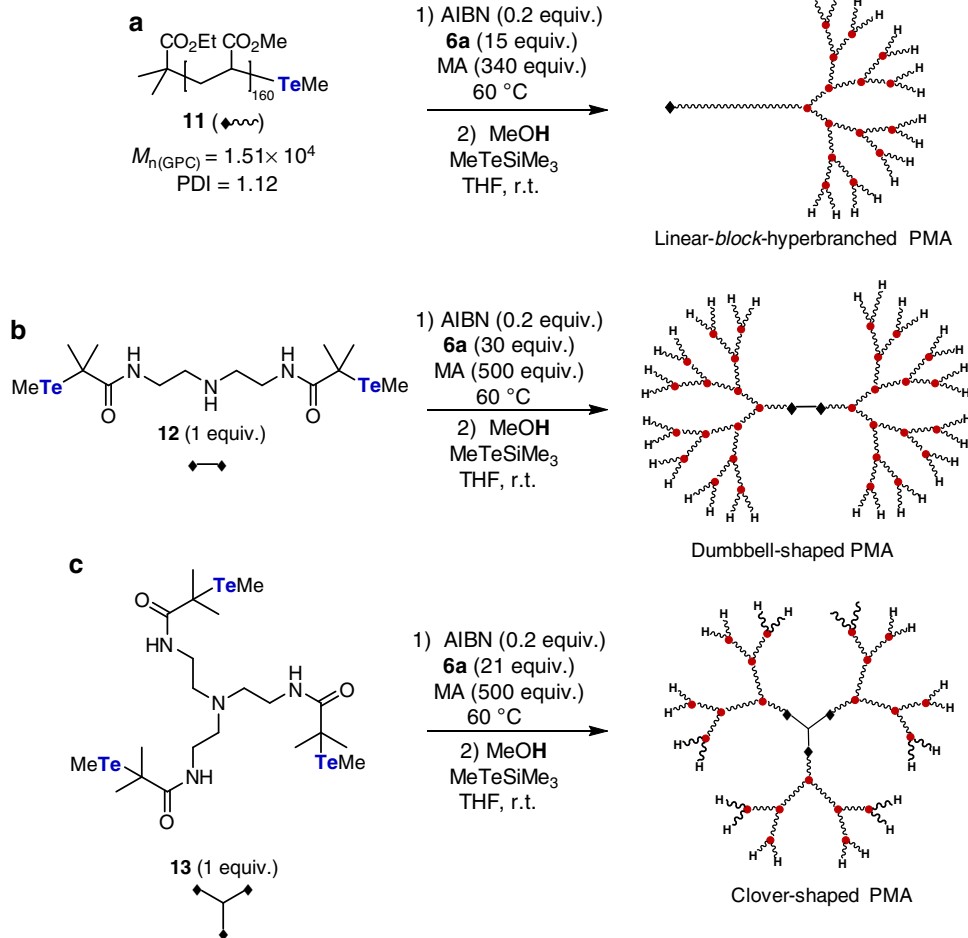

**Fig. 5** Synthesis of hyperbranched PMA with different molecular structures. **a** Linear-*block*-hyperbranched PMA, **b** dumbbell-shaped PMA, and **c** clover-shaped PMA

## Discussion

A method for the synthesis of HBPs with well-controlled structures over the 3D structure was developed through the living chain-growth copolymerization of a monomer with hierarchical reactivity, vinyl telluride. Various HBPs with different 3D structures were synthesized by changing the structure of the CTA and the ratio of the CTA, vinyl telluride, and acrylic monomer. Since this method relies on a living polymerization under radical conditions, a wide range of block copolymers and end-functionalized polymers can be synthesized by using the living ends. In addition, a broad range of functional groups is compatible. Therefore, this work opens an avenue for the design of new polymer structures and functional materials based on HBPs.

## Methods

**Typical procedure for the synthesis of 10a**. A solution of **9** (3.5 μl, 0.02 mmol), **6a** (33.5 μl, 0.30 mmol), MA (0.9 ml, 10 mmol), and AIBN (24 μl in benzene, 0.004 mmol) was heated at 60 °C in the dark for 43 h. The polymerization was quenched by putting the reaction vessel in a freezer at −20 °C for 5 min. The conversions of **6a** (> 99%) and MA (90%) were determined from $^1$H NMR spectroscopy. The crude mixture was dissolved in anhydrous tetrahydrofuran (THF) (5 ml) and CH$_3$OH (140 μl, 3.4 mmol), and Me$_3$SiTeMe (1.18 mol L$^{-1}$ in THF, 320 μl, 0.38 mmol) was added at room temperature. The resulting solution was stirred for 1 h at room temperature. The polymer sample was withdrawn and was analyzed by SEC-MALLS ($M_{n(SEC)}$ = 2.56 × 10$^4$ g mol$^{-1}$, PDI = 1.71, $M_{n(MALLS)}$ = 5.73 × 10$^4$ g mol$^{-1}$).

**Data availability**. The authors declare that the data supporting the findings of this study are available within the paper and its Supplementary Information File. All data are available from the authors on reasonable request.

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

## Acknowledgements

This work was partly supported by a Grant-in-Aid for Scientific Research on Innovative Areas (JSPS KAKENHI, Grant Number JP24109005). We thank Prof. Hiroshi Watanabe, Kyoto University (KU), for stimulating discussions and helpful suggestions as well as Prof. Yasuyuki Nakamura (KU) for assistance with the MALLS measurements.

## Author contributions

S.Y. conceived, designed, and directed the investigations. Y.L. carried out all experiments and analyses, besides the AFM measurements. T.N. and M.T. carried out the AFM measurements. S.Y. and Y.L. wrote the manuscript, and all authors commented on it.

## Additional information

**Competing interests:** The authors declare no competing financial interests.

