## [Peer Review File · Nature Communications]

Reviewers' comments:

Reviewer #1 (Remarks to the Author):

The authors describe a very interesting and intriguing way to introduce very efficiently branching via controlled radical polymerization, more precisely, through a latent chain transfer agent similar as in the RAFT process. Only when a radical is added to the newly designed vinylalkyltelluride compound, the resulting unit having in the "dormant state" two alkyltelluride units starts to act as chain transfer units allowing very efficient branching. The author proved by various spectroscopic experiments, that the branching was very efficient, close to 100%, even though some evidence for back biting was also found.

On average the density of the branching units, or what the authors call "generations", can be well controlled by the ratio of the non-branching CTA and the vinylalkyltelluride, and on average, the distance between the branch points is well controlled by the CTA to monomer ratio. With this intriguing new latent branching CTA unit the authors could continue their successful work on telluride CTAs used for controlled radical polymerization, allowing now also nicely to prepare various branched architecture, e.g. linear/branched block copolymers and dumbbell structures. Certainly, this work fully justifies publication. The work is carried out very carefully and the structures are very well characterized. Also, this latent branching CTA is of high novelty. Still, some criticism has also to be made, especially on the terminology used by the authors, and their claim, that a strong control over the 3D structure of hyperbranched polymers was now achieved, for the first time.

Firstly, the prepared structures are a subclass of the classical hyperbranched structures, and belong to the linear/branched hybrids, since not each monomer can branch out and rather long linear chains are created between the branch points. Such structures have been prepared before, as the authors are aware of, by various other methods, some of them achieving a much better control of the 3D structure (even though prepared e.g. by stepwise anionic or controlled radical polymerization, or by using branched macromonomers). So the term "hyperbranched" is not fully correct for those polymers. In addition, a couple of real hyperbranched polymers have been prepared in one-step process also exhibiting 100% degree of branching (not cited by the authors), certainly, by a fully different concept. Also the term "generations" does not really fit for a one-step process. And the term "stimuli-responsive" for their latent CTA is not fitting at all!

But the most criticism has to be given with regard to the structural control: the resulting structure show dispersities just below 2.0 (around 1.55-2.1). This is far from what can be achieved with real dendrimers or in the reported step-wise synthetic procedures. That means, that certainly features of a controlled polymerization are achieved with control over the growing chain ends and on certain architectures, but in general, the distances of the linear chain between the branch points show a significant dispersity, and also, even though the branching units seem to efficiently branch, there is no control how many monomer unit exactly add before the next branching takes place, just a certain average value can be controlled by the ratio CTA/6a/monomer.

So I fully support publication of this manuscript, but with some revision of the terminology, some update of the references, and not in Nature Communication but a more polymer oriented journal like *Macromolecules* or *Polymer Chemistry*.

In addition, there should be one more revision: Fig.1c shows some model compounds used to calculate the binding energies. In addition, the authors have to give a scheme elucidating exactly the first steps of their branching process showing the full chemical structure of the used CTA, the growing dormant chain and the addition step on 6a. The model compounds 7 and 8 are a bit misleading.

Reviewer #2 (Remarks to the Author):

The paper reports successful synthesis of structurally controlled hyper-branched polymers (HBPs). The idea to use stimuli-responsive monomer (vinyl telluride) is novel. The branching timing is programmed in the monomer, which is a new concept, and the obtained HBPs are structurally very different from the existing HBPs. The paper also demonstrates the wide scope of this synthetic method in terms of monomers, structures (tadpole, dumbbell, and clover), and chain-end functionalities, which may attract interest of material chemists as well as polymer chemists. This work is innovative and the discussion is generally clear. I think that the paper is significant and can be published after minor revision.

(1) Figure 1c: "6" may be replaced by "6a".

(2) Page 5: Why did the polymerization significantly slow down with even a small amount of vinyl telluride? The k_p (propagation rate constant) of the vinyl telluride-chain end propagating radical is so small?

(3) Page 11 line 3: "copolymerization" may be replaced by "copolymerized".

(4) Page 11: What is another possibility (other than back-biting) that the authors would like to exclude by the doubly labeled experiment? This needs explanation.

(5) Table 1: PDI can be smaller with an increase of generation in terms of statistics. The authors may mention this in the text, if they would like to do.

Reviewer #3 (Remarks to the Author):

The authors report on an elegant one-step procedure to generate hyperbranched architectures by means of an organotellurium-mediated living radical copolymerisation reaction of vinyl tellurides with acrylates. The developed method lies in the directionality that can be introduced into the vinyl telluride AB* monomer's reactivity in which (due to their interconnection) the B* functionality only becomes reactive when the A functional end of the monomer has reacted first. With this approach, the step growth mechanism of conventional self-condensing vinyl polymerisations – in which both A and B* functionalities of the monomer do not possess such a distinct difference in reactivity – can be overcome and therefore a better control over the obtained hyperbranched structures (up to the 6th dendritic generation) can be obtained. Moreover, the tellurium endgroup can easily be converted into an amino functionality, whilst different 3D hyperbranched structures could be synthesized by simply changing the structure of the organotellurium-containing CTA and the generality of the approach was demonstrated by using different acrylate containing monomers.

Major comments.

1. The authors describe the vinyl telluride as a stimuli-responsive monomer, a term that is also included in the manuscript title. This stimuli-responsiveness is attributed to the fact that “the monomer does not initiate the polymerization by itself, but it starts the polymerization and acts as a branching point after it has reacted”. The term stimuli-responsive was introduced to describe polymers that are capable to change their chemical and/or physical properties when subjected to an external trigger such as light, temperature, pH, etc.. Hence, the use of “stimuli-responsive monomers” in this work to describe the sequence-defined reactivity nature of the AB* vinyl telluride monomer (i.e. B* only becomes reactive when A has reacted first) is highly confusing, if not incorrect. Its use should therefore be avoided in the title and by extension the entire manuscript. The use of the term “hierarchical reactivity”, introduced by the authors on p4, however is a much better chosen terminology to describe the pre-ordered reactivity of the core monomer.
2. Please address the following remarks with regard to the citations made:
 - A) The two references cited with regard to AB₂ polycondensation reactions (p2) seem rather outdated (i.e. publication year 1952 and 1990). I would thus like to encourage the authors to replace and/or include more recent examples of synthetically advanced AB₂-type procedures (e.g. 10.1021/jacs.5b05060) that better reflect the ongoing improvements related to the field of hyperbranched polymers.
 - B) At least one reference (e.g. 10.1021/cr900068q) should be included when addressing the limited three dimensional control of hyperbranched polymers (p2) to provide readers that are not specialized in the field with potential background information.
 - C) The authors should cite the recent work of Haifeng Gao (10.3390/polym9060188) in which the recent progress on hyperbranched polymers that are synthesised via radical-based self-condensing vinyl polymerisations is reviewed.
3. The dotted line in Figure 2d, representing the theoretical M_n value, reaches approx. 4.25 x 10⁴ g/mol after a 95% conversion of 6a (most right-handed data point). As stated in the text, these results are associated to run 5 depicted in Table 1. However, the theoretical M_n value listed there is 3.46 x 10⁴ g/mol, a contradiction that needs to be addressed. Furthermore, the authors state that the experimental M_n values obtained via MALLS detection are close to the theoretically calculated one, yet the difference between these two values seems to (greatly) extend 20% of the expected molecular weight. For transparency purposes, this should be commented within the manuscript.
4. The authors might consider to elaborate more on the need to control the structure of hyperbranched polymers in view of their final applications. I would also like to have an insight in the limitations and remaining challenges of the proposed methodology that can be targeted in the future to further improve the structural control over the hyperbranched architectures.
5. Last paragraph on page 8: it is stated that the copolymers with different PMA spacer length also give rise to copolymers having a unimodal SEC trace. This statement should be supported with an overview Figure in the Supporting Information, containing the SEC traces of all entries from Table 1, and referred to appropriately.

Minor comments.

1. A cross-reference to all figures and tables should be made within the text, especially when the results thereof are the subject of discussion. In some occasions the wrong figure is referred to, please check and adapt this where needed.

2. At least one reference (e.g. 10.1021/cr900068q) should be included when addressing the limited three dimensional control of hyperbranched polymers (p2) to provide readers that are not specialized in the field with potential background information.
3. p2: "This is due to the presence of step growth mechanism, and not only monomer 1 but also all existing oligomers, such as 2 and 3, and polymers being involved in the propagation step." I would suggest to rephrase, since it is rather difficult to understand the authors' statement.
4. Figure 2c-e: it should be made clear from the caption that these figures show the result of the 6th dendritic generation (run 5).
5. Legend Figure 2c: "VT" is an unexplained abbreviation that appears in the legend for the first time and should be change for "6a".
6. The title of the y-axis in Figure 2d and in of the column heading in Table 1 are annotated as " $M_n \times 10^{-4}$ ". I believe the authors meant to say " $M_n \times 10^4$ ".
7. M_n values throughout the text: please add the appropriate unit, such as g/mol or Da (for example bottom of p11, Figure 5a, Column headings Table 1...).
8. For the readers' understanding, Figure 4d might require a bit more spacing. An increased distance between the structures and text should suffice.
9. The authors might consider to replace the term "GPC" with the IUPAC recommended "SEC" throughout the entire manuscript. Although it is definitely not incorrect to use GPC, the term SEC is believed to better emphasise the method of the underlying separation mechanism (i.e. hydrodynamic volume).
10. No affiliation of the authors is included in the manuscript file.
11. Please give the manuscript a thorough reading, in order to address for the grammatical/spelling errors. Some examples:
 - Hyperbranched: no hyphenate
 - ¹H-NMR, ¹³C-NMR: with hyphenate
 - p2: the requirement; AB₂ monomers; presence of a step growth; specific conditions; intermicellar reactions – p3: stimuli-responsive (Figure 1b, but also check entire manuscript) – p4: B* only participates in the polymerization after A has reacted; cationic polymerization; activation of 7 and 8 in the presence of; CTA 9 at different ratios of; p5: decreased with increasing monomer conversion – p7: leads to (2^N-1) of branching points; after reduction of the methyltellanyl polymer endgroup; *in situ* (italic); (PMMA) standards – p9: which possesses the; polymer molecules; polymer dot was; corresponds to a (caption Fig3) – p10: similar conditions – p11: nearly identical results; Figure 4d – p13: transformed into an amino group; 2-(dimethylamino)ethyl acrylate; compatible with polar functional groups; having narrow PDIs; has been developed; the ratio of; polymers could be synthesized; is compatible with the radical polymerization process; for the design of new polymer...
 - Caption Table 1: no upper cases in the middle of the caption; acrylate monomers...
12. Figure 4b: if a ¹H-NMR spectrum is recorded of 10d-D, an overlay with the ²H-NMR spectrum can be useful to allow for a quick assessment of the branching efficiency, since the resonance peak is expected to not show up in the ¹H-NMR spectra.
13. Figure S2-13 "GPC trace of hyperbranched PMA": provide more detailed information of the hyperbranched structure that is related to the SEC trace.

In conclusion, the current study is interesting and should be published in a leading journal. If it reaches the innovation degree required for Nature Communications can be debated, from a polymer chemistry view point it is certainly very interesting, as the proposed methodology can be an important addition to the toolbox of single monomer methodologies to generate hyperbranched polymer architectures with high control. In any case, the authors need to carefully address all of the above comments. Finally, the manuscript needs to undergo linguistic improvement by a native speaker.

We would like to thank you and the reviewers for the time and efforts in evaluating our manuscript and providing valuable comments. We are pleased to know positive opinions from Reviewers 2 and 3 for the novelty and importance of this work. Although Reviewer 1 expressed negative opinion, we strongly believe that the comments derived from the misunderstanding of our results. We are sending here the revised manuscript according to the reviewers' comments. The revised parts are highlighted by yellow underline. The followings are the point-to-point reply to the comment from reviewers.

Reviewer #1:

Comment. Firstly, the prepared structures are a subclass of the classical hyperbranched structures, and belong to the linear/branched hybrids, since not each monomer can branch out and rather long linear chains are created between the branch points. Such structures have been prepared before, as the authors are aware of, by various other methods, some of them achieving a much better control of the 3D structure (even though prepared e.g. by stepwise anionic or controlled radical polymerization, or by using branched macromonomers). So the term "hyperbranched" is not fully correct for those polymers.

Response. The criticism went to the structure of the hyperbranched polymers we synthesized. While the reviewer pointed out the formation of irregular structure, the kinetic study showing the occurrence of the statistical copolymerization (Figure 2c) and living character (Figure 2e) fully support the formation of regular branching. The formation of "rather long linear chains between the branch points" was also pointed out, but it is incorrect. We agree that "not each monomer can branch out", but the polymer could have rather short chain (~3 monomer units) to create branch points as discussed below. All our results are consistent with the formation of structurally well-controlled and highly symmetrical hyperbranched polymers.

To clarify this point, the average number of monomer units ($\overline{X_{br}}$) inserted between the branched points was discussed in the main text. While we already described the relation among the generation N , the number of branching points, and the number of branched polymer chains in the legend of Figure 2, we moved this part to the main text and added the equation predicting $\overline{X_{br}}$ as eq. 1 (page 5). We also added the calculated $\overline{X_{br}}$ in Table 1, which ranges from 2.9 to 68 depending on the conditions. For example, $\overline{X_{br}}$ s of the 6th and 7th generation polymers using 500 and 2000 equivalent of MA are 2.9 and 7.6 with average branching point of 63 and 127, respectively (Table 1, runs 5 and 10). In addition, we demonstrated to tune $\overline{X_{br}}$ from 3.0 to 60 with keeping the same branching number (Table 1, runs 3 and 7-9). These results clearly illustrated the flexibility and versatility for synthesizing structurally controlled hyperbranched polymers.

The reviewer also comment on the advantage of other method e.g. "by stepwise anionic or controlled radical polymerization", but stepwise synthetic route significantly limits its practical application as we already stated in the main text.

Comment. In addition, a couple of real hyperbranched polymers have been prepared in one-step process also exhibiting 100% degree of branching (not cited by the authors), certainly, by a fully different concept.

Response. We carefully reexamined the literature and found two synthetic methods enabling 100% degree of branching by a one-step process using the AB₂ monomer method by Smet (Angew. Chem. Int.

Ed. 2002, 41, 4547) and Ueda (J. Am. Chem. Soc. 2010, 132, 11000). These works are cited as refs 36 and 37. However, these methods rely on the condensation polymerization and cannot control over three-dimensional structure, i.e., molecular weight, distribution, and branching density. In contrast, our method not only achieved 100% degree of branching but also control over three dimensional structure. Therefore, we believe that just comparing the branching efficiency is inappropriate. During this literature investigation, we also found some papers focusing on the structural control of hyperbranched polymers by using AB₂ monomer method, and these works were cited as Refs. 32-35 with a comment (page 2, line 4 from the bottom). These additions did not affect about the novelty and importance of the current work.

Comment. Also the term “generations” does not really fit for a one-step process.

Response. We agree that conventional hyperbranched polymers cannot define generation because of their unsymmetrical and irregular structure. However, once the structure of hyperbranched polymers becomes controlled and symmetrical like dendrimer, one can define the generation.

Comment. And the term “stimuli-responsive” for their latent CTA is not fitting at all!

Response. Since Reviewer 3 raised the same issue with a specific comment, we will touch this point in the reply to Reviewer 3.

Comment. But the most criticism has to be given with regard to the structural control: the resulting structure show dispersities just below 2.0 (around 1.55-2.1). This is far from what can be achieved with real dendrimers or in the reported step-wise synthetic procedures.

Response. As we already described in the introduction, impact of dendrimers on materials science has been severely limited due to their low availability originated from the requirement of multistep synthesis, though the structural control is perfect (dispersity = 1.00). Our achievement is to realize the synthesis of highly symmetrical hyperbranched polymers in one step, which opens many possibilities in materials design. Therefore, we believe the comment is inappropriate.

Comment. That means, that certainly features of a controlled polymerization are achieved with control over the growing chain ends and on certain architectures, but in general, the distances of the linear chain between the branch points show a significant dispersity, and also, even though the branching units seem to efficiently branch, there is no control how many monomer unit exactly add before the next branching takes place, just a certain average value can be controlled by the ratio CTA/6a/monomer.

Response. As we pointed out above, the kinetic analysis showing the statistical copolymerization (Figure 2c) and living character (Figure 2d) together with the ¹³C labeling experiments clearly reveals that the branching occurs regularly. Furthermore, the growth of the linear polymer chain between the branch points takes place by TERP, which usually gives good control for acrylate polymerization (ref. 11 and 12, dispersity ~ 1.1). While there is no direct method to measure the dispersity of the linear polymer part, the observed experimental results strongly supports controlled structure with narrow dispersity for the linear chains. Therefore, we believe the comment is incorrect.

Comment. So I fully support publication of this manuscript, but with some revision of the terminology, some update of the references, and not in Nature Communication but a more polymer oriented journal like Macromolecules or Polymer Chemistry.

Response. We appreciate the support, and we believe that novelty of the concept and impact for material design are relevant of the scope of Nature Communications. The observed control over three-dimensional structure and versatility shown in this manuscript has never been achieved by a one-pot polymerization reaction. Therefore, a vast number of hyperbranched polymer with new architecture and functional groups would be synthesized by our method. More important, a series of newly designed monomer with hierarchy reactivity would be developed by utilizing this new concept.

Comment. In addition, there should be one more revision: Fig.1c shows some model compounds used to calculate the binding energies. In addition, the authors have to give a scheme elucidating exactly the first steps of their branching process showing the full chemical structure of the used CTA, the growing dormant chain and the addition step on 6a. The model compounds 7 and 8 are a bit misleading.

Response. We thank the reviewer for the suggestion, and we modified Figure 3a and added the intermediate structure for the first branching process as suggested.

Reviewer #2:

Comment. (1) Figure 1c: “6” may be replaced by “6a”.

Response. The typo was corrected as suggested.

Comment. (2) Page 5: Why did the polymerization significantly slow down with even a small amount of vinyl telluride? The k_p (propagation rate constant) of the vinyl telluride-chain end propagating radical is so small?

Response. This is probably due to the slower activation of the dormant species derived from the vinyl telluride (**6a**) than that derived from methyl acrylate, because the formers have higher C-Te bond dissociation energy. We added this point in page 5.

Comment. (3) Page 11 line 3: “copolymerization” may be replaced by “copolymerized”.

Response. We replaced the word as suggested.

Comment. (4) Page 11: What is another possibility (other than back-biting) that the authors would like to exclude by the doubly labeled experiment? This needs explanation.

Response. We just wanted to double check the possibility of back-biting reaction, and the results were consistent with the back-biting. Therefore, we did not add any comment here.

Comment. (5) Table 1: PDI can be smaller with an increase of generation in terms of statistics. The authors may mention this in the text, if they would like to do.

Response. We thank the reviewer for pointing out an important issue. Indeed, we are now elucidating the statistics. However, this is beyond the scope of the current work and will be an important subject of the forthcoming paper.

Reviewer #3:

Comment (major). The authors describe the vinyl telluride as a stimuli-responsive monomer, a term that is also included in the manuscript title. This stimuli-responsiveness is attributed to the fact that “the

monomer does not initiate the polymerization by itself, but it starts the polymerization and acts as a branching point after it has reacted". The term stimuli-responsive was introduced to describe polymers that are capable to change their chemical and/or physical properties when subjected to an external trigger such as light, temperature, pH, etc.. Hence, the use of "stimuli-responsive monomers" in this work to describe the sequence-defined reactivity nature of the AB* vinyl telluride monomer (i.e. B* only becomes reactive when A has reacted first) is highly confusing, if not incorrect. Its use should therefore be avoided in the title and by extension the entire manuscript. The use of the term "hierarchical reactivity", introduced by the authors on p4, however is a much better chosen terminology to describe the pre-ordered reactivity of the core monomer.

Response. We utilized the term "stimuli-responsive monomer" because the property of the monomer (the reactivity of carbon-tellurium bond of the monomer) changes by a stimulus, which is the reaction of polymer-end radical to the monomer. However, as two reviewers pointed out this terminology, we changed the term "stimuli-responsive monomer" to "monomer having hierarchical reactivity" throughout the manuscript including the legend of Figure 1.

Comment (major). Please address the following remarks with regard to the citations made:

A) The two references cited with regard to AB₂ polycondensation reactions (p2) seem rather outdated (i.e. publication year 1952 and 1990). I would thus like to encourage the authors to replace and/or include more recent examples of synthetically advanced AB₂-type procedures (e.g. 10.1021/jacs.5b05060) that better reflect the ongoing improvements related to the field of hyperbranched polymers.

Response. We intended to refer original papers so that we included rather old papers. We looked at the suggested paper, but its main topic is an application of hyperbranched polymers prepared by AB₂ polycondensation reactions. We are more concentrated on methodology for the synthesis of hyperbranched polymer. Therefore, we think this work is inappropriate as a reference. Instead, we found a recent synthetic work using this method (*Angew. Chem. Int. Ed.* 2015, 54, 7631-7635) and this work was included as a reference (ref. 35).

Comment (major). B) At least one reference (e.g. 10.1021/cr900068q) should be included when addressing the limited three dimensional control of hyperbranched polymers (p2) to provide readers that are not specialized in the field with potential background information.

Response. This paper was already cited as Ref. 4.

Comment (major). The authors should cite the recent work of Haifeng Gao (10.3390/polym9060188) in which the recent progress on hyperbranched polymers that are synthesised via radical-based self-condensing vinyl polymerisations is reviewed.

Response. We added this work as Ref. 10.

Comment (major). The dotted line in Figure 2d, representing the theoretical M_n value, reaches approx. 4.25×10^4 g/mol after a 95% conversion of 6a (most right-handed data point). As stated in the text, these results are associated to run 5 depicted in Table 1. However, the theoretical M_n value listed there is 3.46×10^4 g/mol, a contradiction that needs to be addressed. Furthermore, the authors state that the experimental M_n values obtained via MALLS detection are close to the theoretically calculated one, yet the difference between these two values seems to (greatly) extend 20% of the expected molecular weight. For transparency purposes, this should be commented within the manuscript.

Response. We determined Mn(MALLS) from Mw(MALLS) directly obtained by an light scattering experiment and Mw/Mn (PDI) determined from SEC. This is because PDI obtained from MALLS is unreliable due to the low scattering intensity of low-molecular weight region. This may be also the case for Mw(MALLS) so that it was overestimated. We added this point as Ref 42. The difference of Figure 2d and run 5 in table 1 is originated from different calculation method. In Figure 2d, we calculated the molecular weight including TeMe moiety since it is a molecular weight monitoring during the polymerization, while we calculated the molecular weight of the end-hydrogen polymer after reduction for run 5 table 1.

Comment (major). The authors might consider to elaborate more on the need to control the structure of hyperbranched polymers in view of their final applications. I would also like to have an insight in the limitations and remaining challenges of the proposed methodology that can be targeted in the future to further improve the structural control over the hyperbranched architectures.

Response. We agree that this is an important issue. However, we already discussed on the importance of structural control in hyperbranched polymers in the introduction. Also, we are now working on the scope and limitation of this method, and we believe that this point is beyond the scope of the current paper and will be an excellent subject of the next paper. Therefore, we did not change this point.

Comment (major). Last paragraph on page 8: it is stated that the copolymers with different PMA spacer length also give rise to copolymers having a unimodal SEC trace. This statement should be supported with an overview Figure in the Supporting Information, containing the SEC traces of all entries from Table 1, and referred to appropriately.

Response. We appreciate for a valuable suggestion, but an overview figure including all data shown in Table 1 becomes very busy and loses readability. We already shows conceptually same overview figures by using the data presented in Table 1, runs 1-6 (Figures 2f and 2g) and commented the figures in the main text. Therefore, we did not change this point.

Comment (minor). A cross-reference to all figures and tables should be made within the text, especially when the results thereof are the subject of discussion. In some occasions the wrong figure is referred to, please check and adapt this where needed.

Response. We checked very carefully and corrected when necessary.

Comment (minor). At least one reference (e.g. 10.1021/cr900068q) should be included when addressing the limited three dimensional control of hyperbranched polymers (p2) to provide readers that are not specialized in the field with potential background information.

Response. As we stated above, this paper was already cited as Ref. 4.

Comment (minor). p2: "This is due to the presence of step growth mechanism, and not only monomer 1 but also all existing oligomers, such as 2 and 3, and polymers being involved in the propagation step." I would suggest to rephrase, since it is rather difficult to understand the authors' statement.

Response. We modified the sentence from "being involved in the propagation step" to "randomly react each other". I hope this modification increases readability of the text.

Comment (minor). Figure 2c-e: it should be made clear from the caption that these figures show the result

of the 6th dendritic generation (run 5).

Response. We added “for the synthesis of the 6th generation (Table 1, run 5)” in the caption of Figure 2c and 2d as suggested.

Comment (minor). Legend Figure 2c: “VT” is an unexplained abbreviation that appears in the legend for the first time and should be change for “6a”.

Response. We corrected as suggested

Comment (minor). The title of the y-axis in Figure 2d and in of the column heading in Table 1 are annotated as “Mn x 10⁻⁴”. I believe the authors meant to say “Mn x10⁴”.

Response. We corrected as suggested.

Comment (minor). Mn values throughout the text: please add the appropriate unit, such as g/mol or Da (for example bottom of p11, Figure 5a, Column headings Table 1...).

Response. We added g/mol as a unit when it is necessary.

Comment (minor). For the readers’ understanding, Figure 4d might require a bit more spacing. An increased distance between the structures and text should suffice.

Response. We modified Figure 4d as suggested.

Comment (minor). The authors might consider to replace the term “GPC” with the IUPAC recommended “SEC” throughout the entire manuscript. Although it is definitely not incorrect to use GPC, the term SEC is believed to better emphasise the method of the underlying separation mechanism (i.e. hydrodynamic volume).

Response. We changed GPC to SEC as suggested.

Comment (minor). No affiliation of the authors is included in the manuscript file.

Response. We are sorry to forget important information, and we added our affiliation.

Comment (minor). Please give the manuscript a thorough reading, in order to address for the grammatical/spelling errors. Some examples:

- Hyperbranched: no hyphenate
- ¹H-NMR, ¹³C-NMR: with hyphenate
- p2: the requirement; AB2 monomers; presence of a step growth; specific conditions; intermicellar reactions – p3: stimuli-responsive (Figure 1b, but also check entire manuscript) – p4: B* only participates in the polymerization after A has reacted; cationic polymerization; activation of 7 and 8 in the presence of; CTA 9 at different ratios of; p5: decreased with increasing monomer conversion – p7: leads to (2^N-1)-of branching points; after reduction of the methyltellanyl polymer endgroup; *in situ* (italic); (PMMA) standards – p9: which possesses the; polymer molecules; polymer dot was; corresponds to a (caption Fig3) – p10: similar conditions – p11: nearly identical results; Figure 4d – p13: transformed into an amino group; 2-(dimethylamino)ethyl acrylate; compatible with polar functional groups; having narrow PDIs; has been

developed; the ratio of; polymers could be synthesized; is compatible with the radical polymerization process; for the design of new polymer...

- Caption Table 1: no upper cases in the middle of the caption; acrylate monomers... _

Response. We appreciate for pointing out grammatical errors and typos, and we corrected as suggested.

Comment (minor. Figure 4b: if a ^1H -NMR spectrum is recorded of 10d-D, an overlay with the ^2H -NMR spectrum can be useful to allow for a quick assessment of the branching efficiency, since the resonance peak is expected to not show up in the ^1H -NMR spectra.

Response. In the ^1H -NMR spectrum of 10d-D, signals corresponding to the polymer main chain appear at around 2 ppm. Therefore, it is difficult to prove the lack of hydrogen signal directly attached to the ^{13}C carbon by the ^1H -NMR. Therefore, we did not change the Figure.

Comment (minor. Figure S2-13 “GPC trace of hyperbranched PMA”: provide more detailed information of the hyperbranched structure that is related to the SEC trace.

Response. We added the corresponding schematic structure in the Figures as suggested.

Thank you and the reviewers again for kind consideration. We hope that the revised manuscript is now suitable for publication.

REVIEWERS' COMMENTS:

Reviewer #1 (Remarks to the Author):

The authors addressed the issues raised by the reviewers sufficiently and revised the manuscript accordingly.

Just a few typos/grammar errors had been introduced in the newly added manuscript text, so a final editing is advised.

Reviewer #2 (Remarks to the Author):

I think that the paper can be published as is.

Reviewer #3 (Remarks to the Author):

The authors have prepared a revision of their manuscript based on three sets of reviewers' comments. The authors have adequately and carefully engaged with my comments, yet I do see the issues that reviewer #1 has regarding the novelty of the process. Clearly, the science is carefully carried out there is certainly a degree of novelty, yet it is on the authors to make this novelty clear. The actual text has not changed substantially after the revision and I suggest that the authors introduce a section into the text that clearly points out how their study differentiates from the literature. I think in the realm of hyperbranched polymers this is essential, because the field is well established. Clearly, innovations are possible, but the comments of reviewer #1 suggest that it is essential to highlight the novelty and leave no ambiguity.

Finally, I still maintain that the manuscript needs to undergo language editing.

Response to Referees' comments:

The followings are the point-to point reply to the comment from reviewers. Since reviewers 1 and 2 did not pose scientific comments, we response here to the comments from reviewer 3.

Comment. The authors have prepared a revision of their manuscript based on three sets of reviewers' comments. The authors have adequately and carefully engaged with my comments, yet I do see the issues that reviewer #1 has regarding the novelty of the process. Clearly, the science is carefully carried out there is certainly a degree of novelty, yet it is on the authors to make this novelty clear. The actual text has not changed substantially after the revision and I suggest that the authors introduce a section into the text that clearly points out how their study differentiates from the literature. I think in the realm of hyperbranched polymers this is essential, because the field is well established. Clearly, innovations are possible, but the comments of reviewer #1 suggest that it is essential to highlight the novelty and leave no ambiguity.

Reply. This reviewer asked to clarify the novelty of our method over other synthetic routes of the controlled synthesis of hyperbranched polymers. Since this is the first example to use a monomer having hierarchical reactivity, we believe that the novelty the current method is clear. Upon modification of our manuscript from Nature Chemistry format to Nature Communication one, we clearly mentioned the limitation of the previous synthetic method in the abstract (**Several methods have been developed to synthesize HBPs by the one-step procedure under special conditions**). Furthermore, the currently available methods and their limitations are clearly described in the introduction (**page 2, line 1 from the bottom~: HBPs with narrow dispersity were only obtained without any molecular weight and distribution control**).

Comment. Finally, I still maintain that the manuscript needs to undergo language editing.

Reply. We already asked a language editing to a native chemist. We believe that asking the further editing is unnecessary.

Thank you for your support and we hope that the revised manuscript is now suitable for publication.